# Juvenile Huntington’s Disease Skin Fibroblasts Respond with Elevated Parkin Level and Increased Proteasome Activity as a Potential Mechanism to Counterbalance the Pathological Consequences of Mutant Huntingtin Protein

**DOI:** 10.3390/ijms20215338

**Published:** 2019-10-26

**Authors:** Azzam Aladdin, Róbert Király, Pal Boto, Zsolt Regdon, Krisztina Tar

**Affiliations:** 1Department of Medical Chemistry, Faculty of Medicine, University of Debrecen, H-4032 Debrecen, Hungary; aladdin@med.unideb.hu (A.A.); regdon.zsolt@med.unideb.hu (Z.R.); 2Doctoral School of Molecular Medicine, University of Debrecen, H-4032 Debrecen, Hungary; 3Department of Biochemistry and Molecular Biology, Faculty of Medicine, University of Debrecen, H-4032 Debrecen, Hungary; kiralyr@med.unideb.hu; 4Stem Cell Differentiation Laboratory, Department of Biochemistry and Molecular Biology, University of Debrecen, H-4032 Debrecen, Hungary; boto.pal@med.unideb.hu

**Keywords:** Huntington’s disease (HD), juvenile HD fibroblast, mitochondria, proteasome, parkin, cell protection

## Abstract

Huntington’s disease (HD) is an inherited neurodegenerative disorder, caused by an abnormal polyglutamine (polyQ) expansion in the huntingtin protein (Htt). Mitochondrial dysfunction and impairment of the ubiquitin-proteasome system (UPS) are hallmarks of HD neurons. The extraneural manifestations of HD are still unclear. We investigated the crosstalk between mitochondria and proteolytic function in skin fibroblasts from juvenile HD patients. We found reduced mitosis, increased cell size, elevated ROS and increased mitochondrial membrane potential in juvenile HD fibroblasts, while cellular viability was maintained. Mitochondrial OXPHOS analysis did not reveal significant differences compared to control. However, the level of mitochondrial fusion and fission proteins was significantly lower and branching in the mitochondria network was reduced. We hypothesized that juvenile HD fibroblasts counterbalance cellular damage and mitochondrial network deficit with altered proteasome activity to promote cell survival. Our data reveal that juvenile HD fibroblasts exhibit higher proteasome activity, which was associated with elevated gene and protein expression of parkin. Moreover, we demonstrate elevated proteasomal degradation of the mitochondrial fusion protein Mfn1 in diseased cells compared to control cells. Our data suggest that juvenile HD fibroblasts respond to mutant polyQ expansion of Htt with enhanced proteasome activity and faster turnover of specific UPS substrates to protect cells.

## 1. Introduction

Huntington’s disease (HD) is a progressive, incurable, neurodegenerative hereditary disease characterized by the adult onset of motor dysfunctions, psychiatric disturbances, intellectual decline, dementia, and finally death [1]. Early appearance of HD followed by rigidity, dystonia and seizures with more severe and faster progression before 20 years is considered as juvenile HD, which is apparently 8–10% of HD cases [2]. The hallmark of this neurodegenerative disease is intraneuronal inclusions. HD is caused by the expansion (>36) of the polyglutamine (polyQ) repeat in exon 1 of the huntingtin protein, which makes the protein prone to misfolding and subsequently oligomerizing [3]. Monomers of the mutant huntingtin protein polymerize into large aggregates or organize into different types of oligomers with different levels of cellular toxicity. “Toxic” oligomers, of a defined structure or size, are generated before the appearance of visible aggregates [4]. The pathological process is more accelerated in juvenile HD, especially when the CAG repeat number is higher than 60 in the *HTT* gene [2] At the cellular level, HD is associated with defects in transcription, protein turnover, and mitochondria homeostasis, which are characteristic of misfolded protein stress. Despite the large number of studies on HD, no hypothesis clearly describes the pathogenesis of HD. Huntington’s disease has been mainly studied in the central nervous system (CNS). However, the huntingtin protein is also expressed in peripheral tissues [5,6]. Skin primary fibroblasts of adult onset HD patients are an attractive model for studying the disease due to the expanded polyglutamine stretch in the huntingtin protein in these fibroblasts [7]. Several studies describe the alteration of mitochondrial bioenergetics, increased oxidative stress, and changes in gene expression profile in skin fibroblasts derived from adult HD patients [8,9,10,11]. However, mitochondrial dynamics, which is well studied in the neurons of HD [12,13], has not been well elucidated in peripheral tissues in juvenile HD. Imbalanced mitochondrial dynamics is a crucial underlying mechanism for neurotoxicity in Huntington’s disease [14,15,16].

In most eukaryotic cells, mitochondria form a dynamic network and are subject to continuous fission and fusion. Unopposed fission or fusion, in response to the deletion of specific factors, results in a reduction in mitochondrial function [17,18]. The fusion–fission process affects not only the mitochondrial architecture, but also the metabolic status of the cell [19,20,21,22]. The fusion of mitochondria promotes repair and complementation processes, while damaged mitochondria are segregated from the network by fission, promoting selective mitophagy and providing quality control [17,18,23]. Mitochondrial fusion and fission are orchestrated mainly by large GTPases including optic-atrophy 1 (Opa1), mitofusin-1 (Mfn1), and mitofusin-2 (Mfn2) for fusion, and the dynamin-related protein 1 (Drp1) for fission. The fusion–fission process is tightly regulated to maintain balanced mitochondrial dynamics, including the degradation of specific substrates by the ubiquitin-proteasome system (UPS) [24,25,26].

The UPS and autophagy play a crucial role in the maintenance of protein homeostasis through their ability to eliminate damaged and misfolded proteins. UPS and autophagy are vital for numerous cellular processes that are regulated by the temporally specific degradation of pathway components [27,28]. Proteasomal activity is tightly regulated. To promote substrate degradation, the core particle of the proteasome (CP) interacts with proteasome activators, which open the gate and allow specific substrate entry into the core [29,30,31]. 

The UPS is also involved in the regulation of mutant huntingtin aggregation and toxicity. Downregulation of proteasome activity promotes the formation of mutant huntingtin aggregates in both cell and animal models of HD [32]. On the other hand, increasing proteasomal activity with sulphoraphane promotes the elimination of mutant huntingtin in cell culture [33], showing the beneficial effect of increased proteasome activity. The proteasome can eliminate and reduce mutant huntingtin; in contrast, accumulation of toxic huntingtin protein leads to proteolytic collapse and the accumulation of damaged or unneeded proteasome substrates that perturb cellular homeostasis [34,35,36]. These misfolded protein aggregates might also sequester additional proteins, which are essential for cellular homeostasis. 

The crosstalk between the mitochondria and proteolytic machinery is an intensively studied field in many diseases, including neurodegenerative diseases [26,37,38]. However, studies are mainly focusing on the events occurring in the CNS. In the present study, we sought to identify the role of the proteasome and mitochondrial architecture changes at early stages of the disease. Therefore, we used skin primary fibroblasts derived from juvenile HD patients. Skin fibroblasts and cells directly converted from fibroblasts to neurons are currently being used to study HD biology [7,39]. We analyzed two juvenile primary HD fibroblast cell lines. These cell lines were already directly reprogrammed into neurons and the neurons were characterized showing abnormal neurite outgrowth and increased aggregate formation compared to control [40]. In addition, recent studies demonstrated that juvenile HD fibroblasts also showed lower levels of mutant htt protein to wild-type htt protein compared to adult-onset patients [41]. Several clinical studies demonstrate that juvenile HD patients show a broad range of symptoms and signs that only partially overlap with adult-onset HD. Adult-onset presents as a hyperkinetic disorder, while juvenile HD typically presents as a hypokinetic disease. A comprehensive assessment of brain structure of patients with juvenile HD showed that patients with juvenile HD have reduced intracranial volumes, and also the volumes of subcortical regions and of cortical white matter were significantly decreased. The cerebellum however was enlarged. The authors argue that this pattern may explain the unique picture of hypokinetic motor symptoms to the contrary of the hyperkinetic chorea-like phenotype of adult-onset HD. However, the clinical and neuropathological differences between adult-onset and juvenile HD is still not well understood. Another study also argues that the greater recognition of the different clinical manifestations of juvenile HD would facilitate early diagnosis and management of the disease [42,43,44]. We demonstrate higher proteolytic activity and elevated levels of parkin in juvenile HD fibroblasts, which might contribute to cell protection during the early age of the disease. Furthermore, higher proteasome activity promotes specific substrate degradation to counterbalance mitochondrial dysfunction and to maintain mitochondrial quality control.

## 2. Results

### 2.1. Elevated ROS and Increased Mitochondrial Membrane Potential Indicate Alteration of Mitochondrial Function in Fibroblasts from Juvenile HD Patients

Emerging evidence suggests that Huntington’s disease is associated with an increased production of reactive oxygen species (ROS), which participate in neuronal cell death predominantly in the striatum and cortex [45,46,47,48]. Cellular dysfunction in peripheral tissues has not yet been fully studied. Thus, we compared intracellular ROS in skin primary fibroblasts from a healthy (16Q) patient and juvenile HD (68Q and 86Q) patients using carboxy-H2DCFDA. Interestingly, we found significantly elevated cytosolic ROS in fibroblasts with 86 polyQ repeats compared to healthy (16Q) control, which indicates existing oxidative stress in our juvenile HD fibroblast model (Figure 1A). 

We measured intracellular ROS production; however, ROS production in HD neurons predominantly originates from mitochondrial dysfunction [49,50]. Therefore, we also verified mitochondrial status in our cellular models by staining the mitochondria with Mitotracker CXMRos and TMRE. Mitotracker CXMRos is a sensitive indicator of mitochondrial membrane potential changes and oxidative stress, while tetramethylrhodamine ethyl ester (TMRE) loads specifically into polarized mitochondria. In the case of TMRE, we also used a positive control to check the specificity of TMRE. The positive control was generated by treating cells with 20 µM FCCP (Carbonyl cyanide 4-(trifluoromethoxy)phenylhydrazone) to depolarize the mitochondria (Appendix A). FCCP disrupts ATP synthesis by transporting protons across mitochondrial inner membranes. It depolarizes mitochondrial membrane potential and eliminates mitochondrial membrane potential and TMRE staining. Therefore, it can be used as a control to ensure the mitochondrial specificity of the TMRE staining. As shown in Figure 1B,C, we detected a significant increase in mitochondrial membrane potential with both dyes in HD (68Q and 86Q) fibroblasts compared to the healthy (16Q) control. 

### 2.2. Juvenile HD Fibroblasts Demonstrate Reduced Mitochondrial Respiration but Suppressed Glycolysis and Perturbed Cell Cycle

It was previously reported that adult onset HD fibroblasts exhibit slower cellular proliferation compared to their healthy counterparts due to possible alterations in the OXPHOS machinery [10]. We revisited this question and analyzed mitochondrial metabolic activity using Seahorse XF analysis in our cell models. Following normalization to total cellular protein level, mitochondrial activity measurements revealed that HD (68Q and 86Q) fibroblasts exhibited reduced ATP production and basal and maximal respiration. However, the effect did not reach statistical significance (Figure 2A,B). 

Another source of energy production in cellular metabolism is glycolysis. We measured glycolytic activity by monitoring extracellular acidification rate. We detected a significant decrease in glycolysis in 68Q HD fibroblasts compared to healthy (16Q) control (Figure 2C,D). The 86Q HD fibroblasts showed reduced glycolysis compared to healthy (16Q) control, but did not reach significant difference. Taken together, these data suggest that mitochondrial oxidative phosphorylation is maintained while glycolysis is suppressed in juvenile HD fibroblasts providing sufficient energy for cell survival. 

Elevated intracellular ROS, reduced glycolysis, and mitochondrial function can lead to reduced cell viability in neurodegenerative diseases. Thus, we analyzed cell viability/cell death by flow cytometry. FACS analysis show that HD (68Q and 86Q) fibroblasts are viable and that both apoptosis and necrosis are relatively low and comparable to their healthy (16Q) counterpart (Figure 3A).

Cell cycle analysis was performed to examine whether fibroblasts derived from HD patients exhibit cell cycle perturbations as a possible cause for reduced proliferation. We assessed the percentage of cells in G0/G1, S, and G2/M phases. The analysis revealed that the percentages of HD (68Q and 86Q) fibroblasts in S and G2/M phases were significantly lower compared to the healthy (16Q) control, while a significantly higher percentage of HD fibroblasts were in the G0/G1 phase. These data indicate cell accumulation in G0/G1 before the progression of S phase and reduced mitosis in HD fibroblasts (Figure 3B).

In our experiments, we repeatedly observed that HD fibroblasts were larger compared to cells from healthy control. Therefore, we examined cell size/area using high content screening confocal microscopy after phalloidin staining with Texas Red. We found that HD (68Q and 86Q) fibroblasts were significantly larger in cell size (Figure 3C and Appendix A). We concluded that the increased cell size might be associated with the altered cell cycle in fibroblasts derived from juvenile patients with HD.

### 2.3. Juvenile HD Fibroblasts Show Altered Mitochondrial Fusion–Fission Proteins Expression

Mitochondrial function is also maintained by the mitochondrial fusion–fission process. Increased mitochondrial membrane potential might be associated with the altered fusion–fission machinery [20,51,52,53]. Thus, we explored the changes in mRNA levels for the mitochondrial fusion protein genes, *OPA1* (Opa1), *MFN1* (Mitofusin-1), and *MFN2* (Mitofusin-2), and the mitochondrial fission protein genes, *MIEF1* (Mid51), *MIEF2* (Mid49), *FIS1* (Fis1), *DNM1L* (Drp1), and *MFF* (Mff), in HD fibroblasts. We found that the mRNA expression of the mitochondrial fission protein Drp1 was significantly lower in HD fibroblasts expressing the 86 polyQ repeats relative to the healthy control (Figure 4A). 

We also evaluated protein levels for a subset of mitochondrial fusion–fission proteins by Western blot, and by fluorescent confocal microscopy for Drp1. Interestingly, our results revealed that Drp1 protein level was lower in both models of HD (68Q and 86Q) fibroblasts compared to the control (16Q) and reached significance in 86Q similar to the mRNA expression data (**B**,**C**), while Mff did not show significant changes (**D**). Protein levels for the mitochondrial fusion proteins, Opa1 and Mfn1 were significantly lower in HD fibroblasts compared to the control (**E**–**G**). Taken together, the data suggest that the mitochondrial fusion–fission machinery is affected in our model of HD fibroblasts.

Next, we explored whether we see morphological changes in the mitochondria using high content screening confocal microscopy (Figure 5A). 

Quantitative assessment of mitochondrial morphology was performed using live cell high content image analysis with Mitotracker Red CMXRos to stain the mitochondria. We found significantly decreased mitochondrial branching in fibroblasts from juvenile HD patients (68Q and 86Q) compared to the healthy control (16Q) (Figure 5B). Concomitantly, we also found that mitochondrial length was significantly shorter in HD fibroblasts compared to the healthy control (Figure 5C).

### 2.4. Higher Proteasome Activity but Similar Autophagy Rate in Juvenile HD Fibroblasts

We hypothesized that juvenile HD fibroblasts counterbalance the mitochondrial structural deficit to promote cell survival with altered proteasome activity. Ubiquitin-proteasome impairment is well-established in neurons and neuronal models of HD [32,34,54], but not in peripheral tissues. We studied the relative distribution of proteasome complexes in unfractionated cell lysates after native gel separation followed by an in-gel activity assay in the presence and absence of MG132, a reversible proteasome inhibitor. In a native gel, three major bands are usually detected: the core particle (20S), the core particle associated with one regulatory particle (26S), and the core particle associated with two regulatory particles (30S) [55,56]. The characteristic migration of these species can be seen in Figure 6A. We are also able to visualize the activity of the sole core particle, 26S, and 30S by incubating the native gel with a specific LLVY-AMC fluorogenic substrate. The substrate is hydrolyzed by the chymotryptic-like activity of the proteasome and the signal can be visualized under UV. The fluorogenic, proteasome-specific Suc-LLVY-AMC assay revealed higher proteasome activity for both doubly (30S) and singly (26S) capped proteasomes (Figure 6A) in HD fibroblasts compared to control. 

To confirm equal protein loading, we also determined the protein level of the proteasome using an antibody for the β1 subunit of the 20*S* by Western blot. As shown in Figure 6B, the β1 subunit of the 20*S* proteasome protein level in both control and HD fibroblasts were comparable, suggesting higher proteasomal proteolytic activity in the cells derived from HD patients.

The 20*S* proteasome capped with the 19*S* regulatory particle(s), called the 26*S* for singly and 30*S* for doubly capped proteasomes, mediate ATP- and ubiquitin-dependent substrate degradation. As shown in Figure 6A, we detected higher chymotryptic-like activity of the 26*S* and the 30*S* proteasomes in HD patient-derived fibroblasts. Therefore, we investigated the ubiquitination of proteins from total cell lysates. Ubiquitinated protein levels were significantly higher in HD fibroblasts following proteasome inhibition by MG132 compared to the healthy control (Figure 6C). 

Autophagy is the other important recycling process to eliminate damaged or long-lived proteins. Thus, we checked the level of ubiquitinated proteins following autophagy inhibition with bafilomycin A1. We found that inhibition of autophagy slightly increased the ubiquitinated protein pool in HD fibroblasts similar to healthy control (Figure 6D). Using Western blot analysis, we measured protein levels for the autophagy markers, LC3 I and II and p62, in the presence or absence of bafilomycin A1. LC3 II is required for the formation of autophagosomes and can be used as a quantitative marker of autophagy [57] We found that bafilomycin A1 similarly and significantly increased the level of LC3 II in both, HD fibroblasts and healthy control compared to their respective vehicle treated control (Figure 6E). As a substrate for autophagy, p62 is also a marker for autophagy activity [58]. Western blot analyses revealed that autophagy is active and does not show major alteration in 68Q HD fibroblasts compared to healthy control according to the similar accumulation of p62 following inhibition of autophagy by bafilomycin A1 (Figure 6F). As for 86Q HD fibroblasts, we detected active autophagy pathway, but significantly less p62 following autophagy inhibition compared to the bafilomycin treated healthy control (Figure 6E,F). 

These observations demonstrate that the HD fibroblasts in our model exhibit higher proteasome capacity. Furthermore, inhibition of the activity of the proteasome leads to elevated accumulation of ubiquitinated proteins.

### 2.5. Accelerated Mfn1 Degradation Might Be Associated with Elevated Parkin Level in Juvenile HD Fibroblasts

Parkin is an E3 ubiquitin ligase with protective cellular functions. Parkin is also involved in proteasomal-mediated protein turnover, mitochondrial function, and cell survival [59,60,61]. We speculated that the increased level of ubiquitinated proteins following proteasome inhibition and maintained cell viability in HD fibroblasts might also be associated with elevated parkin levels. We investigated the mRNA and protein expression level of parkin in control and juvenile HD fibroblasts. Our data indicate elevated mRNA in both HD and significantly higher mRNA in 68Q fibroblasts and significantly higher protein level in both HD (68Q and 86Q) fibroblasts compared to the control (16Q) (Figure 7A,B). 

We detected a reduced level of mitochondrial fission–fusion proteins (Figure 4), including Mfn1 in HD fibroblasts. It is known that parkin ubiquitinates and targets Mfn1 for proteasomal degradation in human cells [62]. Thus, we utilized a CHX chase assay to determine Mfn1 half-life in healthy and HD fibroblasts in the presence and absence of MG132. The addition of lethal doses of the translation inhibitor CHX blocked new protein synthesis. Thus, the degradation of a protein in vivo can be monitored. We observed elevated degradation of Mfn1 in juvenile HD fibroblasts over 18 h (Figure 7C), suggesting increased proteasome activity and more efficient turnover of Mfn1, in agreement with our results shown in Figure 4F.

In contrast, we found no evidence for proteasome-mediated degradation of Drp1 or Opa1 in the presence or absence of proteasome inhibition in both, control and HD fibroblasts (Appendix A).

## 3. Discussion

Several studies describing a link between mitochondrial quality control and the proteolytic machinery in models of neurodegenerative diseases have been published [26,37,38]. However, these models mainly include studies on post-mortem tissues, in vitro neuronal cells, non-neuron relevant cellular models, and non-human, particularly mouse and rat primary cells. HD is a neurodegenerative disease; however, alteration of cellular homeostasis is observed in peripheral tissues as well [6,7,8,9,10,63,64]. Thus, studying peripheral tissues in HD might add additional details for the development of biomarkers of the disease or model system for drug testing. We used skin fibroblasts derived from patients with juvenile HD to better understand the connection between mitochondrial dysfunction and protein quality control, and to investigate early signs of altered cell homeostasis. Early studies in adult-onset HD fibroblasts demonstrated oxidative stress which is very often accompanied by metabolic alterations [9,10]. Several studies described the relationship between ROS production and mitochondrial membrane potential [65]. Mitochondria produces more ROS at higher membrane potential [66,67] compared to normal conditions. On the contrary, in certain mitochondrial disorders, the membrane potential is decreased and the respiratory chain complexes are dysfunctional, but ROS production is increased [68]. Here, we demonstrate that juvenile HD fibroblasts exhibit higher ROS levels indicating cellular stress condition, which is accompanied by a significantly increased mitochondrial membrane potential measured by two different dyes. Many bioenergetics studies in different models expressing mutant huntingtin suggest that ATP production is decreased and mitochondrial membrane potential is lost [69,70]. ATP synthesis is strictly regulated by the mitochondrial membrane potential, which is maintained by the mitochondrial respiratory chain enzymatic activity. A study conducted by del Hoyo et al. [9] demonstrated that adult-onset HD skin fibroblasts only show reduced catalase activity, but other enzymes related to respiratory chain activity or oxidative stress were not altered compared to their healthy control. In our study, we show that, following normalization of OCR to total protein content, juvenile HD fibroblasts did not produce significantly less ATP and as mentioned mitochondrial membrane potential was significantly elevated. Thus, the maintenance of ATP concentration with the increased mitochondrial membrane potential might be one of the first compensatory responses against the presence of misfolded mutant htt protein in juvenile HD fibroblasts. 

Mitochondrial changes are often associated with imbalanced mitochondrial fusion–fission. Emerging data demonstrate that the relationship between mitochondrial bioenergetics and mitochondrial network organization is bidirectional, suggesting an important role of the fusion–fission process in the maintenance of optimal mitochondrial function [17,19,21]. Mitochondria go through a frequent fusion and fission process. To address whether we see changes in the key mitochondrial proteins responsible for the fusion–fission events, we studied mRNA and protein levels by qRT-PCR and Western blot, respectively. In addition, to visualize mitochondrial network and quantify mitochondrial length, we applied high content screening confocal microscopy. According to a recent study, fusion is a brief process and triggers fission [71]. Increased mitochondrial membrane potential assumes high probability of subsequent fusion, while decreased membrane potential is accompanied by a reduced probability of fusion. We found that juvenile HD fibroblasts show a marked decrease in the fusion proteins Opa1, Mfn1 and Mfn2 and decreased mitochondrial branching. However, the fission protein Drp1 was also reduced at both mRNA and protein levels, while Mff was unaltered in juvenile HD fibroblasts compared to control. It is tempting to speculate that the expression of mutant huntingtin protein in juvenile HD fibroblasts provokes oxidative stress accompanied by an increase in mitochondrial membrane potential to produce ATP and to promote cell survival by removing misfolded proteins. The increased mitochondrial membrane potential would initiate subsequent mitochondrial fusion. However, because of the reduced protein pool, fusion is limited. Furthermore, the decreased level of Drp1 suggests that the fission process is also deficient. Nevertheless, why do we see reduced protein level of mitochondrial fusion–fission GTPases in juvenile HD fibroblasts? The *HTT* gene is ubiquitously expressed and the involvement of the mutant huntingtin is reported to be involved in the expression of specific genes [11]. Here, we show that the expression of *DNM1L*, the gene coding for Drp1 protein, is down regulated in juvenile HD fibroblasts and the protein level is also significantly reduced. Since the gene expression of the mitochondrial fusion proteins were either not affected or did not change significantly, we speculate that cells eliminate fusion proteins to counteract downregulation of Drp1 leading to elevated mitochondrial fragmentation.

Hence, we measured increased chymotryptic- like proteasome activity in juvenile HD fibroblasts and our data confirm such idea. Impairment of the ubiquitin-proteasome system is a hallmark of HD in brain and in *in vitro* neuronal cell models [4,35]. Overexpression and gene therapy of the proteasome activator PA28γ in HD neuronal models enhances cell survival providing neuroprotection and improves motor coordination [72,73]. The presence of the pathological huntingtin protein leads to elevated ROS production and to oxidative stress, not only in neuronal cell models but also in HD skin fibroblasts. Furthermore, the presence of misfolded proteins can activate the proteasome and, in turn, the elevated proteasome capacity results in improved clearance of specific substrate proteins. We demonstrate that proteasome inhibition in juvenile HD fibroblasts leads to increased accumulation of ubiquitinated proteins. In addition, the mitochondrial fusion protein Mfn1, which is a known substrate for proteasomal degradation, shows an enhanced turnover in our cell model. Parkin is a diverse ubiquitin ligase and the type of ubiquitination that parkin is able to promote will determine the degradation pathway of the substrate. Several studies suggest that parkin may regulate mitochondrial dynamics by inhibiting mitochondrial fusion via promoting ubiquitination and enhanced degradation of Mfn1 by the proteasome [74,75]. In our study, Mfn1, but not Drp1 or Opa1, was shown to require parkin for its ubiquitination and subsequent degradation by the proteasome. Endogenous Mfn1, in addition, requires mitochondrial damage to induce its parkin-dependent degradation [62]. We demonstrate here that juvenile HD fibroblasts exhibit an elevated expression of parkin on both the mRNA and protein levels. Furthermore, the ubiquitination of proteins was significantly higher in juvenile HD fibroblasts after inhibition of the proteasome, while autophagy activity was not significantly affected. That suggests that parkin might also modulate the activity of the 26S proteasome in our cell model by a similar mechanism already published by Um et al. [60]. Further studies are required however, to demonstrate how parkin regulates autophagy in juvenile HD fibroblasts.

Collectively, the data presented here suggest that mitochondrial dynamics is affected in juvenile HD fibroblasts. The presence of the pathological huntingtin protein results in imbalanced mitochondrial fusion–fission. Drp1 expression is down regulated and, to counteract, cells promote the degradation of Mfn1 by enhancing the activity of the proteasome and increasing the expression of parkin. On the other hand, the elimination process needs energy in the form of ATP for proteasome function. To cover the energy demand, the mitochondria generate ATP, which is normally accompanied by mitochondrial fusion. In our model, the enhanced degradation of Mfn1, however, might lead to the inhibition of fusion machinery. In addition, the higher mitochondrial membrane potential and higher ROS production cause oxidative stress, which acts back on mitochondrial homeostasis. Furthermore, we see suppressed glycolysis and this metabolic dysfunction might precede the defects in the respiratory chain itself at later age of the disease.

Our results argue that certain events of the disease are significantly different at the juvenile age, including altered mitochondrial dynamics, elevated parkin level, and increased proteasome activity in HD skin fibroblasts. This should be taken into account when interpreting results from juvenile HD models. Finally, finding a mechanism to maintain the activity of the protein clearance pathway and to restore the mitochondrial network in neurons might provide a strategy to alleviate the disease at the adult age.

## 4. Materials and Methods 

All materials were purchased from Sigma-Aldrich (St. Louis, MO, USA) unless specified otherwise.

### 4.1. Ethics Statement

The following cell lines were obtained from the NIGMS Human Genetic Cell Repository at the Coriell Institute for Medical Research: AG07095, GM04281, and GM05539 (Camden, NJ, USA). The Coriell Institute maintains the written consent forms and privacy of the donors of the fibroblast samples, and the authors had no contact or interaction with the donors. All human fibroblast cells and protocols in the present study were carried out in accordance with the guidelines approved by the University of Debrecen (approval date, 10-09-2018).

### 4.2. Cell Culture

One control juvenile skin fibroblast cell line (AG07095) with 16 polyQ [40] and two diseased cell lines; GM04281 with repeat number CAG 1: 71 and CAG 2: 17, and GM05539 with repeat number CAG 1: 97 and CAG 2: 22 [40,41], (Appendix A) from juvenile HD patients were used in our study. Fibroblasts were maintained in minimum essential medium (MEM), supplemented with 10% non-heat inactivated fetal bovine serum (FBS, Gibco™, Thermo Fisher, Waltham, MA, USA), 2 mM L-glutamine, 100 units/mL penicillin, and 100 µg/mL streptomycin, which is termed complete MEM (CMEM), at 5% CO_2_ and 37 °C. All experiments were performed on fibroblasts with comparable passage numbers, ranging from 7 to 14, in order to avoid possible effects of cellular senescence.

### 4.3. Reactive Oxygen Species (ROS) Measurement

Fibroblasts were seeded in 6-well plates at the density 10^5^ cells per well. On the following day, cells were incubated with 1 μM Carboxy-H2DCFDA (Thermo Fisher) in CMEM with reduced FBS (2%) for 30 min at 5% CO_2_ and 37 °C. Cells were harvested by Trypsin-EDTA, rinsed with 1× PBS and resuspended in 1× PBS complemented with 1% (*v*/*v*) FBS. 6-carboxy-2’,7’-dichlorodihydrofluorescein diacetate is a chemically reduced, acetylated form of fluorescein used as an indicator for reactive oxygen species (ROS) in cells. This nonfluorescent molecule is readily converted to a green-fluorescent form when the acetate groups are removed by intracellular esterases and oxidation (by the activity of ROS) occurs within the cell.

Cells were filtered through a 41 µm Nylon Net Filter (Merck Millipore, MilliporeSigma, Burlington, MA, USA prior to the FACS measurement. Data acquisition was performed using a FACS Aria III flow cytometer (BD Biosciences, Franklin Lakes, NJ, USA). Data were analyzed with FlowJo software version 10 (FlowJo LLC, Ashland, OR, USA). 

### 4.4. Mitochondrial Membrane Potential Measurement

Mitochondrial membrane potential (ΔΨm) measurement was performed by two different staining methods. 

### 4.5. A. Mitotracker Red CMXRos 

Fibroblasts seeded in 6-well plates at the density 10^5^ cells per well. On the following day, the media was removed, cells were rinsed with serum-free CMEM and were incubated with 50 nM Mitotracker Red CMXRos (Thermo Fisher) in serum-free CMEM at 37 °C and 5% CO_2_ for 30 min.

Following incubation, cells were harvested by Trypsin-EDTA, washed twice with sterile 1× PBS and resuspended in 200 µL 1× PBS complemented with 1.0% (*v*/*v*) FBS. Cells were filtered through a 41 µm Nylon Net Filter (Merck Millipore) prior to the FACS measurement.

Data acquisition was performed using a FACS Aria III flow cytometer (BD Biosciences). Data were analyzed with FlowJo software.

### 4.6. B. Tetramethylrhodamine Ethyl Ester (TMRE) Staining

Cells were treated with and without 20 µM of Carbonyl cyanide-4-(trifluoromethoxy) phenylhydrazone (FCCP) (Abcam, Cambridge, UK) for 10 min prior the 100 nM TMRE staining (Abcam) in CMEM at 5% CO_2_ and 37 °C for 10 min. Following incubation cells were harvested by Trypsin-EDTA, washed twice with sterile 1× PBS and resuspended in 200 µL 1× PBS complemented with 1.0% (*v*/*v*) FBS. Cells were filtered through a 41 µm Nylon Net Filter (Merck Millipore) prior to the FACS measurement. Data acquisition was performed using a FACS Aria III flow cytometer (BD Biosciences). Data were analyzed with FlowJo software.

### 4.7. Seahorse XF Analysis 

Healthy and HD fibroblasts were seeded (2 × 10^4^ cell/well) in CMEM in a XF96 cell culture microplates (Seahorse Bioscience, Billerica North Billerica, MA, USA) with the appropriate background correction wells and incubated overnight at 5% CO_2_ and 37 °C. In parallel, the sensor cartridge was prepared by adding 200 µL of Seahorse Bioscience XF96 calibrant solution (pH 7.4) (Part No: 100-840-000, Seahorse Bioscience) to each well of a Seahorse Bioscience 96-well utility plate. The sensors with the calibrant solution were incubated overnight at 37 °C without CO_2_. The measurement was performed using Seahorse XF96 Analyzer. For XF Cell Mito Stress analysis, the following day, CMEM was replaced with 180 µL of XF assay medium (Seahorse Bioscience) supplemented with 2 mΜ L-glutamine (XFL) and 1 g/L glucose, and then the plate was incubated at 37 °C without CO_2_ for 1 h. After 20 min equilibration time, oxygen consumption rate (OCR) was measured every 6 min (1 min mixing, 5 min measurement) for five loops. The mitochondrial inhibitors were applied at the final concentrations as follows: 1 µM oligomycin (O), 1 µM FCCP (F), 1 µM antimycin-A (A), and 1 µM rotenone (R). 

For XF Glycolysis stress, CMEM was replaced with 180 µL XFL media and incubated for 1 h at 37 °C without CO_2_. After equilibration, ECAR (extracellular acidification rate) was obtained every 9 min (1 min mixing, 5 min measure, 3 min wait) for five loops after the addition of the following compounds: 10 mΜ glucose (G), 1 μΜ oligomycin (O), and 50 mΜ 2-deoxy-D-glucose (2-DG). 

Bradford protein assay was used to measure protein concentration for normalization. Data analysis was performed with Wave 2.3 Agilent Seahorse Desktop software.

### 4.8. Cell Cycle Analysis

Healthy and HD fibroblasts were seeded in CMEM (10^5^ cells/per well). Cell cycle analysis was performed by flow cytometry according to the protocol from Abcam (https://www.abcam.com/protocols/flow-cytometric-analysis-of-cell-cycle-with-propidium-iodide-dna-staining) with slight modification using 10 µg/mL propidium-iodide (Sigma). Data acquisition was performed using a FACS Aria III flow cytometer (BD Biosciences). Data were analyzed with FlowJo software.

### 4.9. Apoptosis Assay

Annexin V-FITC Apoptosis Detection Kit I (BD Pharmingen™) was used according to the manufacturer’s instructions. Data acquisition was performed by FACS Aria III flow cytometer (BD Biosciences, USA). Data were analyzed with FlowJo_V10 software.

### 4.10. Immunofluorescence Staining 

Healthy and HD fibroblasts were grown on cover slips coated with 1% gelatin at the density of 2 × 10^4^ cell/well. To stain the mitochondria, the media was removed, and the cells were incubated at 37 °C in a 5% CO_2_ incubator with 50 nM Mitotracker Red CMXRos (M7512, Thermo Fisher) in serum-free media for 20 min. After staining, the cells were carefully rinsed twice with 1× PBS, fixed for 15 min with 3.7% paraformaldehyde (PFA), and rinsed again twice with 1× PBS. 

Cells were permeabilized with 0.1% Triton X-100 in PBS for 35min at RT, were incubated for 1 h in blocking buffer (3% BSA; 0.01% Triton X-100 in PBS). Primary antibody for Drp1 was applied for 90 min at RT at 1:1000 dilution. Cells were washed three times with TPBS, and then incubated for 90 min with the secondary antibody Alexa Fluor 488 (Themo Fisher) and with 1 µg/mL DAPI. Coverslips were mounted on slides with Mowiol 4-88: Dabco 33-LV (1:50). Cells were imaged using a Leica SP8 confocal laser scanning microscope.

### 4.11. High Content Screening

For cell size measurement, cells were seeded in cell carrier-96 ultra microplates (Perkin Elmer, Waltham, MA, USA) at the density of 1000 cell/well in CMEM. After 48 h, the cells were washed and fixed for 15 min in 4% PFA and permeabilized with 0.1% Triton X-100 for 15 min at RT. Cells were incubated for 30 min in blocking buffer (1% BSA in PBS). Then, 2 U/mL Texas Red™-X Phalloidin and 1 µg/mL DAPI were applied for 1 h to stain F-actin and the nuclei, respectively. Images were taken with an Opera Phenix High Content Screening System (Perkin Elmer). A 10× air objective (Na 0.3) was used in non-confocal mode using the DAPI (405/435–480) and Alexa 568 (561/570–630) channels. Image analysis was performed with the built in Harmony software. Images were segmented based on DAPI and Alexa 568 channels to detect nucleus and cytoplasm, respectively. Border touching objects were removed and the cell area was calculated.

To quantify mitochondrial species cells were seeded in cell carrier-96 ultra microplates (Perkin Elmer) at the density of 2000 cell/well in CMEM. After 20 h, the media was removed and cells were rinsed with serum-free CMEM. Cells were incubated at 5% CO_2_ and 37 °C, for 30 min with 50 nM Mitotracker Red CMXRos (Thermo Fisher) and 10 µM Hoechst in serum-free CMEM. Cells were washed once, and then covered with 100 µL of FluoroBrite phenol-red-free DMEM medium (Gibco) supplemented with 1% (*v*/*v*) FBS and 2 mΜ l-glutamine. Live cells were subjected to confocal imaging using an automated confocal microscopy Opera Phenix High-Content Screening System (Perkin Elmer).

To detect mitochondrial morphology, Z stacked images were taken with 40× water objective (Na 1.1) in confocal mode with 1 µm distance between a total of 4 stacks. Channels for Hoechst and Mitotracker Orange were used. Mitochondria were analyzed on a maximum projected image. First nuclei was detected with Hoechst channel then mitochondria with the “Find Spots” building block. The readout parameters were spots/nuclei and average spot length.

### 4.12. RNA Extraction and cDNA Reverse Transcription

Total RNA was extracted using TRI Reagent (Molecular Research Center, Inc, Cincinnati, OH, USA) following the manufacturer’s protocol. cDNA synthesis was performed with High-Capacity cDNA Reverse Transcription Kit (Applied Biosystems, Foster City, CA, USA), 1 µg of total RNA was used to reverse transcribe with random primers. 

### 4.13. Quantitative Real-Time PCR

Quantitative real-time PCR (qRT-PCR, was performed with the LightCycler 480 thermocycler (Roche, Basel, Switzerland) using SYBR Green (Xceed qPCR probe, Institute of Applied Biotechnologies, IAB, Strašnice, Czech Republic) according to the manufacturer’s protocol. The primers used in this study are provided in the Appendix A.

### 4.14. SDS-PAGE and Western Blot

The cells were rinsed with 1× PBS and lysed in RIPA buffer (50 mM Tris-HCl, 150 mMNaCl, 0.5% Na-deoxycholate, 2 mM EDTA, 1% NP-40, and 50 mM NaF) supplemented with protease inhibitor cocktail (1 mM PMSF, 1 mM benzamidine and 1× EDTA-free protease inhibitor cocktail:cOmplete tablets Mini- EDTA-free, Roche). Bradford assay was used to determine protein concentration (Quick Start Bradford Protein Assay, Bio-Rad) according to the manufacturer’s instructions. Total cell lysates (20 µg protein/well) were separated by SDS-PAGE, then transferred to nitrocellulose membranes (0.45 µm NCAmersham. GE Healthcare Life Sciences, Chicago, IL, USA) and were subjected to immunodetection. Western Blotting Luminol Reagent (#sc-2048, Santa Cruz Biotechnology, Dallas, TX, USA was used to visualize the membrane-bound peroxidase. The images were taken using a ChemiDoc Imager and signal intensity was analyzed by Image Lab software. The antibody list is provided in the Appendix A. Primary antibodies are listed in Appendix A.

### 4.15. Cycloheximide (CHX) Chase Assay

Cells were seeded in 6 cm dish at the density 2 × 10^5^. On the following day, cells were treated with 300 µg/mL CHX at various time points from 0 to 18 h. To block proteasome activity, cells were preincubated with 10 μM MG132 for 1 h prior to the CHX treatment. Cells were harvested, lysed and Western blotting was performed as described.

### 4.16. Proteasome Activity Assay

Proteasome activity in cell extracts was measured based on the hydrolysis of fluorogenic substrate Suc-LLVY-AMC (Enzo Life Sciences, Farmingdale, NY, USA) for chymotrypsin-like peptidase activity in the presence and absence of 10 µM MG132. All buffers were filtered through a 0.2 µm filter; and each step was performed at 4 °C. The cell pellets were mechanically homogenized for 1 h on ice in lysis buffer (20 mM HEPES, 5 mM MgCl_2_, 320 mM sucrose, 0.2% NP-40, 2 mM EDTA, and 1 mM ATP) supplemented with protease inhibitor cocktail (Roche) and 10 μΜ MG132 was added to the lysis buffer. Homogenates were centrifuged under 16,000× *g* for 10 min at 4 °C. Protein concentrations were measured by Bradford assay. 50 µg protein from each extract in loading buffer (50 mM Tris-HCl, pH 7.4, 5 mM MgCl_2_, 6 % glycerol, and 12 ng/mL xylene cyanol) were loaded on native gel for proteasome activity in loading buffer.

Electrophoresis was carried out in running buffer (90 mM Tris, 90 mM boric acid, 5 mM MgCl_2_, 0.5 mM EDTA, pH 8.3, and 0.5 mM ATP). The applied voltage was 60 V for 12 h at 4 °C. 

Following electrophoresis, native gels were incubated with reaction buffer (50 mM Tris-HCl, pH 7.4, 5 mM MgCl_2_, 1 mM ATP, and 100 µM Suc-LLVY-AMC in DMF) for 30 min at 30 °C in dark. Fluorescent bands were visualized and quantified by ChemiDoc system and bands intensity were analyzed using Image Lab software v4.1(Bio_Rad Lab., Hercules, CA, USA). Proteasome activity were normalized to protein amount. 

### 4.17. Statistical Analysis

Data from each experiment were summarized with the mean and standard deviation (SD) of *n* ≥ 3 experiments. Statistical analysis was performed with ANOVA test (*n* > 2 cell lines analysis) and t-test (*n* = 2 cell lines analysis) using Graphpad Prism 8.2.1. Statistical significance was determined as * *p* < 0.05, ** *p* < 0.01, *** *p* < 0.001, **** *p* < 0.0001.

## Figures and Tables

**Figure 1 ijms-20-05338-f001:**
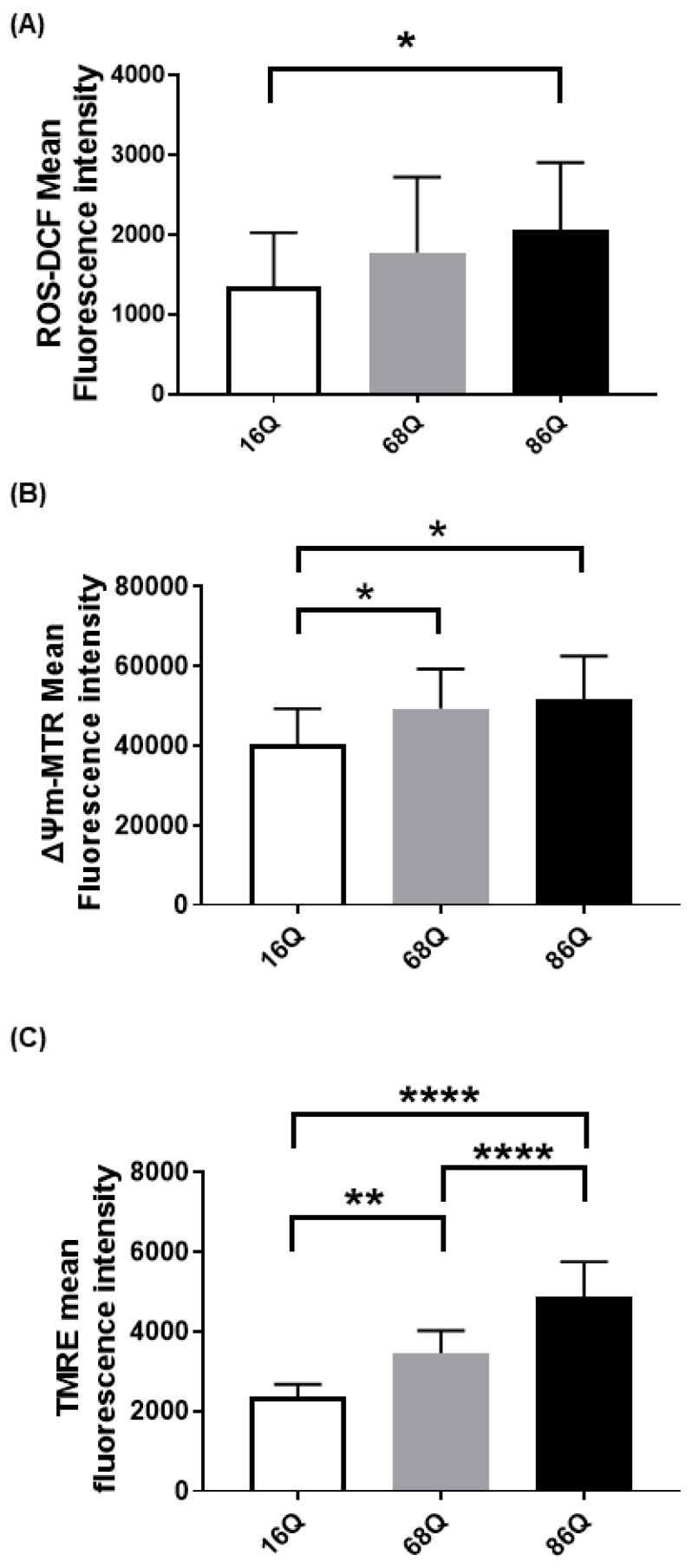
Elevated reactive oxygen species (ROS) production and increased mitochondrial membrane potential in HD fibroblasts. (**A**) ROS levels were determined by staining cells with 1 µM of Carboxy-H2DCFDA for 30 min and followed by flow cytometry. Results are displayed as mean ± SD of five independent experiments and statistically significant differences were calculated using ANOVA test by GraphPad Prism 8.2.1 (* *p* < 0.05, ** *p* < 0.01, *** *p* < 0.001). Mitochondrial membrane potentials of HD (68Q and 86Q) and control (16Q) were assessed using: Mitotracker Red CMXRos (**B**); and TMRE (**C**). Data are presented as mean values ± SD for four independent experiments. Data were analyzed using FlowJo_V10. Statistical analysis was performed using ANOVA test by GraphPad Prism 8.2.1 (* *p* < 0.05; ** *p* < 0.01, **** *p* < 0.0001 compared to healthy control).

**Figure 2 ijms-20-05338-f002:**
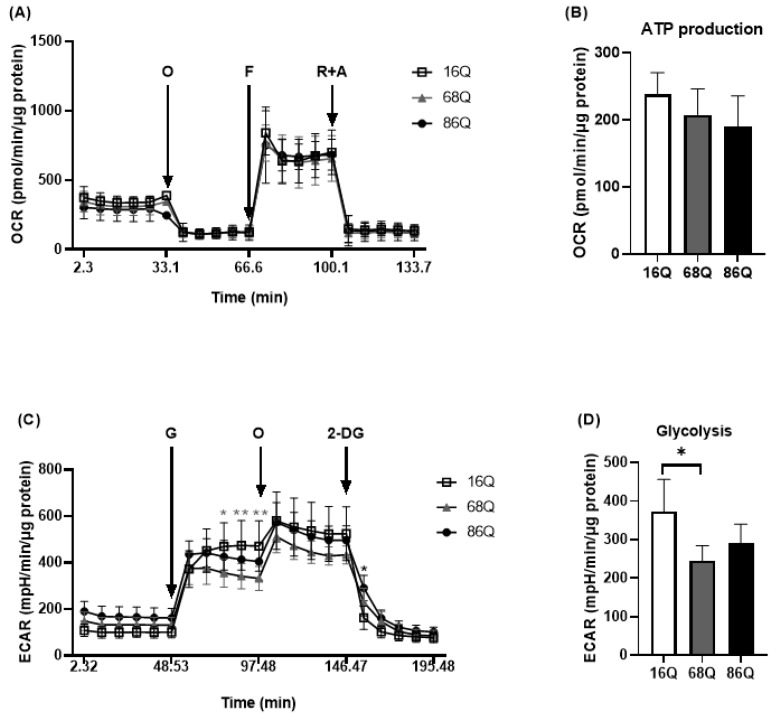
HD patients’ derived fibroblasts show reduced mitochondrial and glycolytic activity. The bioenergetics profile of mitochondrial activity and glycolytic assay were performed using Seahorse XF 96 analyzer. (**A**,**B**) HD (68Q and 86Q) and control (16Q) fibroblasts were seeded in an XF96 cell culture microplate at 20,000 cell/well for 24 h. The following day the basal OCR was determined for 30 min prior to the injection of Oligomycin (1 µM, O); Carbonylcyanide-4-(trifluoromethoxy) phenylhydrazone (1 µM, F); and Rotenone (1 μM, R) with Antimycin A (1 µM, A). (**A**) Bioenergetics profile of mitochondrial function in juvenile HD (68Q and 86Q) and control (16Q) fibroblasts. (**B**) The bioenergetics profile of mitochondrial ATP production. Data are presented as mean values ± SD for three independent experiments. Data were analyzed using Wave Desktop software. Statistical analysis was performed using ANOVA test by GraphPad Prism 8.2.1 (* *p* < 0.05; ** *p* < 0.01, *** *p* < 0.001 **** *p* < 0.0001 compared to healthy control). OCR data were normalized to protein content (pmol/min/µg protein). (**C**,**D**) Glycolytic function was assessed by measuring extracellular acidification rate (ECAR). HD (68Q and 86Q) and control (16Q) fibroblasts were seeded in an XF96 cell culture microplate at 20,000 cell/well for 24 h. The following day, the media was replaced with glucose-free media and cells were incubated for 1 h without CO_2._ The basal ECAR was determined for 30 min in glucose-free medium prior to the injection of 10 mM glucose (G), 1 µM olygomycin (O), and 50 mM 2-deoxy-d-glucose (2-DG) (**C**) Glycolytic profile of HD (68Q and 86Q) and control (16Q) fibroblasts. (**D**) Measure of glycolysis as the ECAR rate after the addition of saturating amounts of glucose production. Data are presented as mean values ± SD for four independent experiments. Data were analyzed using Wave Desktop software. Statistical analysis was performed using ANOVA test by GraphPad Prism 8.2.1 (* indicates *p* < 0.05, ** indicates *p* < 0.01 compared to healthy control). ECAR data were normalized to protein content (pmol/min/µg protein).

**Figure 3 ijms-20-05338-f003:**
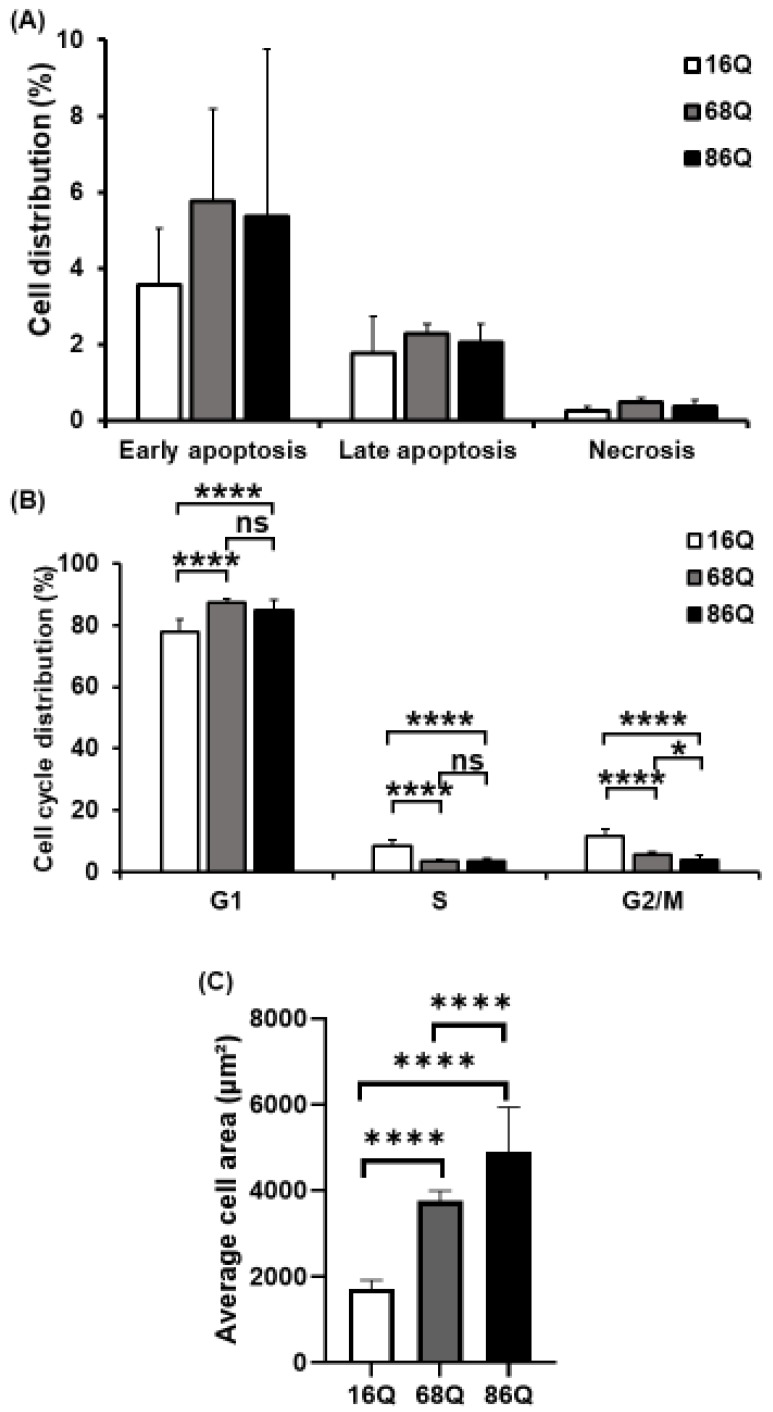
Decreased mitotic rate and increased cell size of juvenile HD fibroblasts. (**A**) Assessment of cell viability by flow cytometry using Annexin-V-FITC and PI. The results are presented as the mean ± SD of three independent experiments. Cell lines were compared using ANOVA test by GraphPad Prism 8.2.1 (* indicates *p* < 0.05, ** indicates *p* < 0.01, *** indicates *p* < 0.001 as compared to control). (**B**) Cell cycle distributions were measured by flow cytometry 24 h after synchronized cell seeding and staining cells with 10 µg/mL propidium-iodide (PI). Results are displayed as mean ± SD of four independent experiments. Statistically significant differences were calculated using ANOVA test by GraphPad Prism 8.2.1 (* indicates *p* < 0.05, ** indicates *p* < 0.01, *** indicates *p* < 0.001). (**C**) Cell area in µm^2^ was determined by staining HD (68Q and 86Q) and healthy (16Q) fibroblasts with Texas-Red Phalloidin for cytoskeletal F-actin, and DAPI for cell nuclei. High content screening (HCS) measurement and quantification were performed by using the Opera Phenix high content screening system. Statistically significant differences were calculated using ANOVA test by GraphPad Prism 8.2.1. Data represent the mean ± SD, *n* = 5 (* *p* < 0.05; ** *p* < 0.01, *** *p* < 0.001 **** *p* < 0.0001).

**Figure 4 ijms-20-05338-f004:**
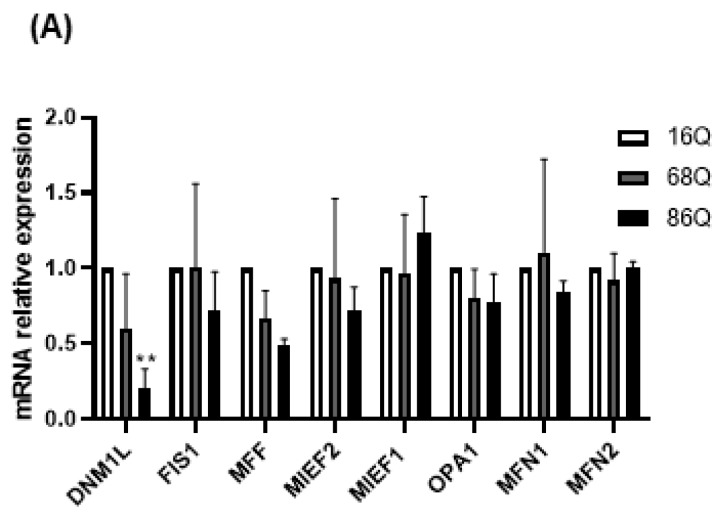
Altered gene and protein expression profile of the mitochondrial fusion–fission machinery in HD fibroblasts. (**A**) The mRNA levels of mitochondrial fission and fusion genes were examined by qPCR. The mRNA levels of HD (68Q and 86Q) fibroblasts were normalized to the healthy (16Q) control. Data are presented as mean values ± SD and represent three independent experiments. Statistical analysis was performed using ANOVA test by GraphPad Prism 8.2.1 (* indicates *p* < 0.05, ** indicates *p* < 0.01, *** indicates *p* < 0.001). (**B**) Representative, merged immunofluorescence confocal images showing the mitochondrial structure using 50 nM Mitotracker Red CMXRos, the mitochondrial fission protein Drp1 with an anti-Drp1 antibody/Alexa Fluor 488 and DAPI to stain the cell nuclei in healthy and juvenile HD fibroblasts. (**C**–**G**) Levels of mitochondrial fusion and fission proteins were examined by SDS-PAGE and Western blot analyses. Protein (20 µg) from total cell lysates were loaded onto SDS-PAGE and transferred to nitrocellulose membrane for Western blot analyses. Actin was used as an internal loading control. Normalized mitochondrial fusion–fission protein levels in HD (68Q and 86Q) fibroblasts were compared to the healthy (16Q) control. Data are presented as mean values ± SD and represent three independent experiments. Statistical analysis was performed using ANOVA test by GraphPad Prism 8.2.1 (* *p* < 0.05; ** *p* < 0.01, *** *p* < 0.001 **** *p* < 0.0001).

**Figure 5 ijms-20-05338-f005:**
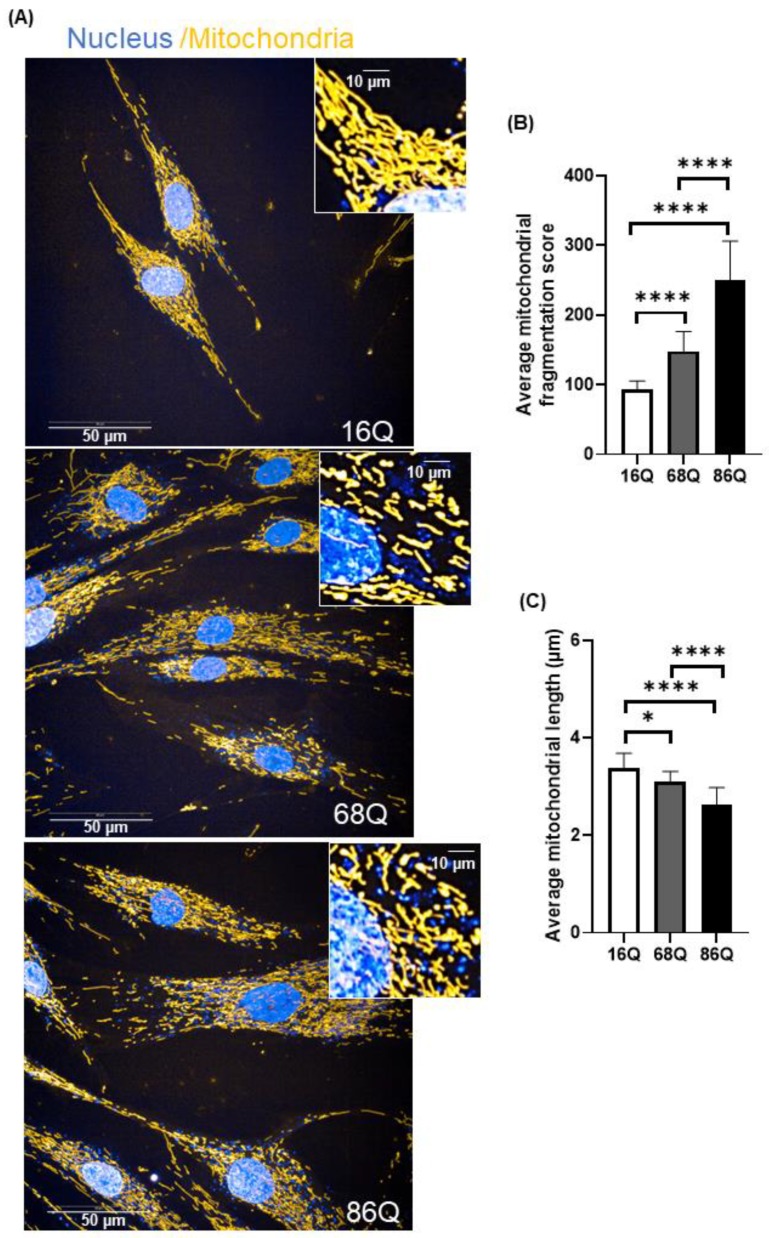
Increased mitochondrial fragmentation and reduced mitochondrial length in juvenile HD fibroblasts are not associated with reduced viability. (**A**) Representative HCS confocal fluorescent imaging of mitochondrial morphology in healthy (16Q) and juvenile HD (68Q, and 86Q) fibroblasts using 50 nM Mitotracker Red CMXRos. (**B**,**C**) Quantification of mitochondrial population and measurement of mitochondrial length in healthy and HD fibroblasts were performed using image analysis with the built in Harmony software. Two thousand cells were analyzed per condition. Data are presented as mean values ± SD and represent five independent experiments. Statistical analysis was performed using ANOVA test by GraphPad Prism 8.2.1 (* *p* < 0.05; ** *p* < 0.01, *** *p* < 0.001 **** *p* < 0.0001).

**Figure 6 ijms-20-05338-f006:**
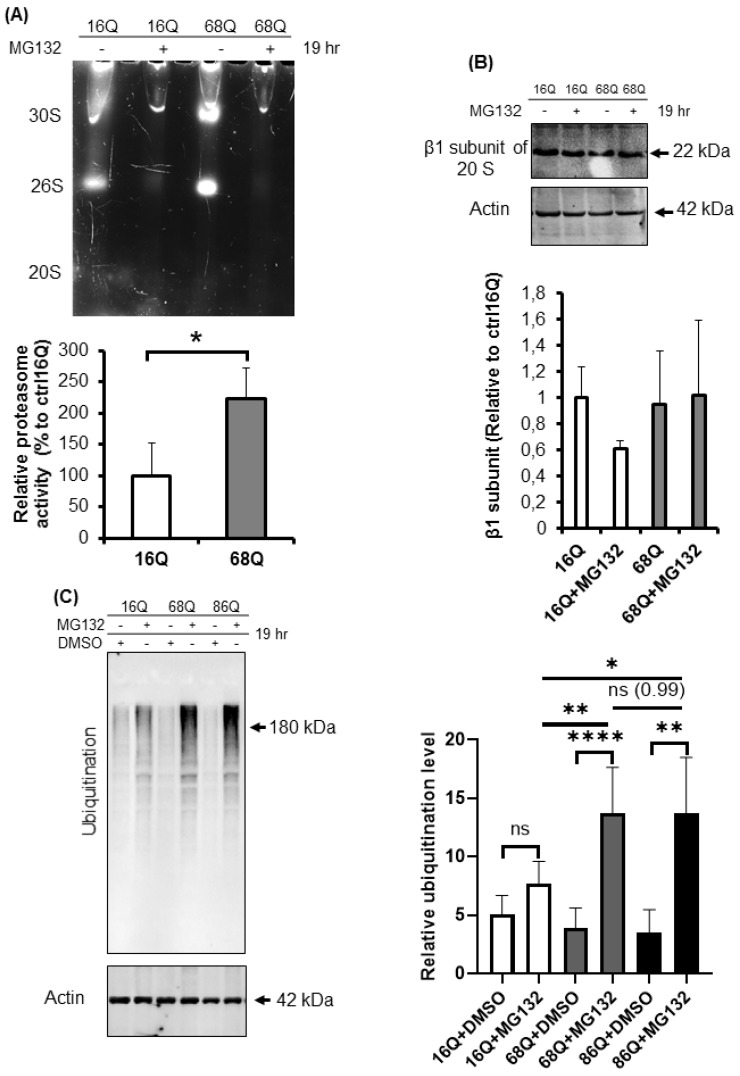
Proteolytic activity and ubiquitination in HD and healthy fibroblasts. (**A**) Proteasome complex activity were analyzed by 3–8% native gel electrophoresis in both healthy and HD fibroblasts. Cells were incubated in the presence and absence of 10 μΜ MG132 proteasome inhibitor and were assayed with the fluorogenic Suc-LLVY-AMC proteasome substrate (upper panel). The activity of the proteasome in juvenile HD fibroblasts was normalized to healthy control. Data represent the mean ± SD, *n* = 3 independent experiments (lower panel). (**B**) Whole cell lysates (50 µg total protein) were loaded onto SDS-PAGE and transferred to nitrocellulose membrane for Western blot analyses to assess proteasome protein level in the presence and absence of 10 µM MG132 using an antibody for the proteasome core β1 subunit. Actin was used as an internal loading control (upper panel). Quantification of proteasome β1 subunit was performed from three independent experiments. Protein levels in HD fibroblasts were normalized to healthy control. Data represent the mean ± SD, *n* = 3 independent experiments (lower panel). (**C**) Whole cell lysates (10 µg total protein) were loaded onto SDS-PAGE and transferred to nitrocellulose membrane for Western blot analyses to assess ubiquitination in the presence and absence of 10 µM MG132. Actin was used as a loading control (upper panel). Protein levels in HD fibroblasts were compared to healthy control. Data represent the mean ± SD, *n* = 3 independent experiments (lower panel). (**D**) Whole cell lysates (10 µg total protein) were loaded onto SDS-PAGE and transferred to nitrocellulose membrane for Western blot analyses to assess ubiquitination in the presence and absence of 100 nM Bafilomycin 1 (BMA1). Actin was used as a loading control (upper panel). Protein levels in juvenile HD fibroblasts were compared to healthy control. Data represent the mean ± SD, *n* = 3 independent experiments (lower panel). (**E**,**F**) Western blot analyses of autophagy markers LC3I, LC3 II and p62 in healthy and HD fibroblasts. Whole cell lysates (20 µg total protein) were loaded onto SDS-PAGE and transferred to nitrocellulose membrane for Western blot analyses to assess autophagy activity in the presence and absence of 100 nM Bafilomycin 1 (BMA1). Actin was used as a loading control (upper panels). Protein levels in HD fibroblasts were compared to healthy control. Data represent the mean ± SD, *n* = 3 independent experiments (lower panels). (* *p* < 0.05; ** *p* < 0.01, *** *p* < 0.001 **** *p* < 0.0001).

**Figure 7 ijms-20-05338-f007:**
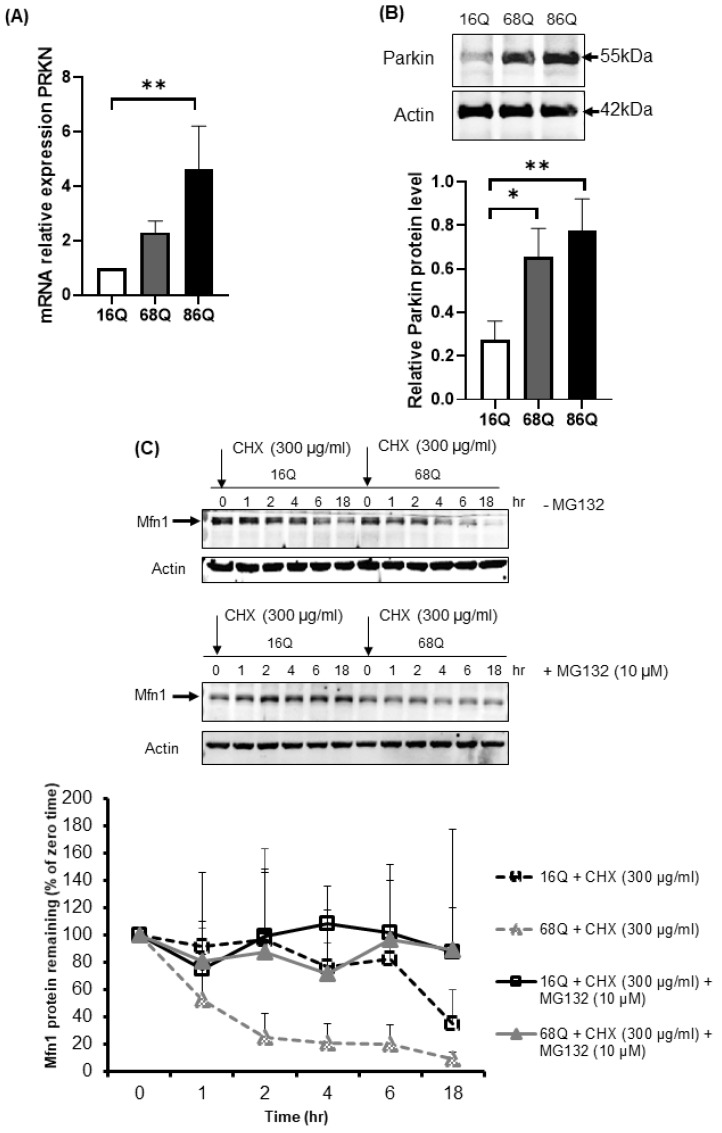
Upregulation of both mRNA and protein levels for parkin in juvenile HD fibroblasts accelerate Mfn1 degradation. (**A**) The mRNA level of PRKN (parkin) was examined by qPCR. The data of mRNA levels of HD (68Q and 86Q) fibroblasts were normalized to the healthy control (16Q). Data are presented as mean values ± SD and represent three independent experiments. Statistical analysis was performed using ANOVA test by GraphPad Prism 8.2.1 (* indicates *p* < 0.05, ** indicates *p* < 0.01, *** indicates *p* < 0.001). (**B**) Equal protein amounts were subjected to SDS-PAGE followed by immunodetection with a Parkin-specific antiserum. Actin was used as loading control. Protein levels in HD (68Q and 86Q) fibroblasts were compared to healthy control (16Q). Data represent the mean ± SD, *n* = 3 independent experiments (lower panels). Statistical analysis was performed using ANOVA test by GraphPad Prism 8.2.1 (* indicates *p* < 0.05, ** indicates *p* < 0.01, *** indicates *p* < 0.001). (**C**) Mfn1 turnover was determined in healthy and HD fibroblasts in the presence and absence of 10 µM MG132. New protein synthesis was blocked by 300 µg/mL CHX. At the time points indicated, cells were harvested and lysed, and equal protein amounts were subjected to SDS-PAGE followed by immunodetection with an Mfn1-specific antibody. Actin was used as loading control (upper panel). Chemiluminescence intensities of the bands in four independent experiments were quantified using the Image Lab software, and corrected for the loading control (actin) (lower panel). Statistical analysis was performed using student t-test by GrapPad Prism 8.2.1 (* indicates *p* < 0.05, ** indicates *p* < 0.01, *** indicates *p* < 0.001).

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
