# Peer review of "Juvenile Huntington’s Disease Skin Fibroblasts Respond with Elevated Parkin Level and Increased Proteasome Activity as a Potential Mechanism to Counterbalance the Pathological Consequences of Mutant Huntingtin Protein"

_ijms, 2019, doi:10.3390/ijms20215338_

Round 1
Reviewer 1 Report
In this study the Authors investigated the crosstalk between mitochondria and proteolytic function in skin fibroblasts from 2 juvenile HD patients as compared to 1 juvanile healthy control. Reduced mitosis and mitochondrial fusion-fission proteins and branching, and increased cell size, ROS and mitochondrial membrane potential were found in HD fibroblasts. Unchanged cell viability and mitochondrial OXPHOS were reported. Increased proteasome activity, parkin expression and Mfn1 degradation were also observed in HD cells.
This study is of interest in the research field, results are clearly presented, methods sound good and it is well written. Anyway, the principal concern about this manuscript derives from the fact that all results and conclusions are based on the comparison between 2 patient cell lines and only 1 control cell line. Considering the individual variability that occurs in subject-derived cells, the use of 1 control cell line strongly limits the value and reliability of obtained results. Considering that obtaining skin fibroblasts from healthy subjects cannot be considered as an objective difficulty, all obtained results should be verified using an adequate number of control cell lines.
Reviewer 2 Report
The study is about phenotypic analysis of the reprogrammed juvenile HD fibroblast and mechanistic analysis of mitochondrial and proteolytic dysfunctions. In this manuscript, A. Aladdin et al., provide convincing evidence that the juvenile HD fibroflasts exhibited increase of proteaosome activity and UPS turnover rate. The study is novel and I feel that this study can have an impact in the field, but I have some concerns that need to be addressed prior to the suitability of this manuscript for publication:
Major Comments:
Authors have provided information (Line 96) about the difference between juvenile and adult HD fibroblast. Any more clinical information about the differences between juvenile and adult HD in terms of the symptoms or phenotypes i.e severity, no matter in CNS or PNS. These may provide more solid rationale on the importance of this study, besides lack of study as a rationale. Any rationale of not include 86Q in Fig 6. I would encourage authors to include this cell line in the analysis too. Did author measure the levels of wildtype, mutant HTT and Oligomers in 16Q control and 68Q,86Q HD cell lines? Any difference between the control and HD cell lines. or this has been provided in recent research article? It would be important to provide this information as it is an hallmark of HD.Minor Comments:
Most of the methods were described properly. However, authors should provide details on statistical analyses. Line 127: What does FCCP stand for? Fig 2D, was the difference between 86Q and 16Q significant? Only 68Q was labeled a significant difference. However, Line160-162:" We detected a significant decrease in glycolysis in HD (68Q and 86Q) fibroblasts compared to healthy (16Q) control (Fig2C and D)." Please clarify. Please provide references for statement "Mitochondrial function is also maintained by the mitochondrial fusion-fission process. Increased mitochondrial membrane potential might be associated with the altered fusion-fission machinery." (Line 193-195) It would be better to provide higher resolution images for Fig 5A. Would the author clarify a bit more for the results as shown in Fig 6A. What did the native gel result indicate? Would the author provide p value for analyses Fig 6D-F between BMA1-treated 16Q and 68Q. The graphs show reduction of protein levels in 68Q especially Fig 6F shows a significant difference between 16Q and 68Q. However, Author stated that "Western blot analyses showed that autophagy is active and comparable in both healthy298 and HD fibroblasts (Fig6D and E)." (Line 297-298)
Reviewer 3 Report
I enjoyed reading this interesting work. The introduction is well written and give a clear context to this study. The authors extensively evaluated HD fibroblast using different approaches. The major finding is the accumulation of Parkin in the 68Q and 86Q fibroblast. However, the main problem of this study is the statistical analysis.
These are my comments and suggestions to validate their findings and conclusions.
They should perform an ANOVA analysis when they are comparing 3 cell lines. Perform this stat analysis where is applicable. On Fig1. A. They should calculate the rate of DCF oxidation on the first 15 min, which will inform whether is more production of ROS in the HD fibroblasts. Section 2.3 requires format. On Fig. 4. The Western blots do not match the quantification. Same for Fig.5A. For example, by "eye", Parkin is increased 5-10 fold in 68Q and 86Q fibroblasts. On Fig. 5. Show higher magnifications. On Fig. 6. Why did not show data of 86Q cell line. Same comment for Fig. 7C. On Fig. 7C. There are not SD in the graph. There is not Stat section on the methods.
Round 2
Reviewer 1 Report
No other request.Reviewer 2 Report
Most of the comments have been carefully responded and addressed. I have no further comment.
Reviewer 3 Report
The author addressed my queries. They still have to double check the English.